# β-Lactoglobulin-Modified Mesoporous Silica Nanoparticles: A Promising Carrier for the Targeted Delivery of Fenbendazole into Prostate Cancer Cells

**DOI:** 10.3390/pharmaceutics14040884

**Published:** 2022-04-18

**Authors:** Maedeh Koohi Moftakhari Esfahani, Seyed Ebrahim Alavi, Peter J. Cabot, Nazrul Islam, Emad L. Izake

**Affiliations:** 1School of Chemistry and Physics, Faculty of Science, Queensland University of Technology (QUT), 2 George Street, Brisbane, QLD 4000, Australia; maedeh.koohi@hdr.qut.edu.au; 2Centre for Materials Science, Queensland University of Technology (QUT), 2 George Street, Brisbane, QLD 4000, Australia; 3School of Medicine and Dentistry, Griffith University, Gold Coast, QLD 4215, Australia; ebrahim.alavi@griffithuni.edu.au; 4School of Pharmacy, The University of Queensland, Woolloongabba, QLD 4102, Australia; pcabot@pharmacy.uq.edu.au; 5School of Clinical Sciences, Faculty of Health, Queensland University of Technology, 2 George Street, Brisbane, QLD 4000, Australia; nazrul.islam@qut.edu.au; 6Centre for Immunology and Infection Control (CIIC), Queensland University of Technology, Brisbane, QLD 4000, Australia

**Keywords:** anthelmintic, fenbendazole, mesoporous silica nanoparticle, prostate cancer

## Abstract

The clinical utilization of fenbendazole (FBZ) as a potential anticancer drug has been limited due to its low water solubility, which causes its low bioavailability. The development of a drug nanoformulation that includes the solubilizing agent as a drug carrier can improve solubility and bioavailability. In this study, Mobil Composition of Matter Number 48 (MCM-48) nanoparticles were synthesized and functionalized with succinylated β-lactoglobulin (BLG) to prevent early-burst drug release. The BLG-modified amine-functionalized MCM-48 (MCM-BLG) nanoparticles were loaded with FBZ to produce the drug nanoformulation (FBZ-MCM-BLG) and improved the water solubility and, consequently, its anticancer effects against human prostate cancer PC-3 cells. The prepared FBZ-MCM-BLG was characterized in terms of size, zeta potential, drug loading capacity, morphology, thermal and chemical analyses, drug release, cellular uptake, cell viability, cell proliferation, production of reactive oxygen species (ROS), and cell migration. The results demonstrated that the FBZ-MCM-BLG nanoparticles have a spherical morphology with a size and zeta potential of 369 ± 28 nm and 28 ± 0.4 mV, respectively. The drug loading efficiency of the new nanoformulation was 19%. The release of FBZ was pH-dependent; a maximum cumulative release of about 76 and 62% in 12 h and a burst release of 53 and 38% in the first 0.5 h was observed at pH 1.2 and 6.8, respectively. The prepared FBZ-MCM-BLG formulation demonstrated higher cytotoxicity effects against PC-3 cells by 5.6- and 1.8-fold, respectively, when compared to FBZ and FBZ-MCM nanoparticles. The new formulation also increased the production of ROS by 1.6- and 1.2-fold and inhibited the migration of PC-3 cells when compared to the FBZ and FBZ-MCM nanoparticles, respectively. Overall, FBZ-MCM-BLG nanoparticles improved FBZ delivery to PC-3 cells and have the potential to be evaluated for the treatment of prostate cancer following a comprehensive in vivo study.

## 1. Introduction

Fenbendazole (FBZ) is an anthelmintic that has been widely studied as a repurposed drug for the treatment of cancer [1,2,3]. FBZ functions through the prevention of tubulin polymerization, thereby impeding cell mitosis [4]. It is used orally as an antiparasitic agent [5]; however, its delivery is limited due to the aggressive environment of the gastrointestinal tract (GIT; pH range of 1.5–8), as well as poor stability, solubility, and permeability [6].

Drug loading into porous nanoparticles has been recently proposed to address some of the challenges of drug delivery, such as low solubility and permeability of drugs [7]. Mesoporous silica nanoparticles (MSNs) have been demonstrated as promising drug carriers due to their high porosity and controlled particle and pore size, as well their ability to chemically modify their molecular structure with hydroxyl, amino, thiol, polyethylene glycol (PEG), and carboxyl functional groups [6,8,9,10]. For example, Mobil Composition of Matter No. 48 (MCM-48)-based nanoparticles have a high surface area and pore volume of ~1000 m^2^/g and ~1 cm^3^/g, respectively. MCM-48 possess a three-dimensional ordered porous network with a pore size of ~3 nm, which is sufficient to accommodate small drug molecules, such as FBZ [6,11]. MCM-48 nanoparticles can increase drug solubility and permeability due to the change of its physical state from crystalline to amorphous when it is loaded in the carrier nanopores [7]. MCM-48 nanoparticles demonstrate high chemical, temperature, and mechanical stability and are biocompatible compounds with high drug loading capacity and no premature release [12].

Because drugs must be solubilized and permeated through the intestinal barrier, pH-responsiveness proteins have also been proposed to partially solve the challenges of oral drug delivery [13]. Proteins are natural biodegradable materials that can be modified with chemical reactions, such as succinylation and acetylation, to improve their solubility and surface hydrophobicity [14,15]. β-lactoglobulin (BLG), a low-cost whey protein, has been demonstrated to acquire good emulsifying and gelation properties after modification using the succinylation reaction. This modification decreases the protein solubility at acidic pH (pH < 5) and increases it at pH > 5.

Thus, in this study, we aimed to load FBZ into succinylated BLG-modified, amine-functionalized MCM-48 nanoparticles (FBZ-MCM-BLG) to protect FBZ throughout GIT and improve its anticancer effects against prostate cancer PC-3 cells. In addition, the release of FBZ from FBZ-loaded MCM nanoparticles (FBZ-MCM) and FBZ-MCM-BLG at both pH 1.2 and 7.4 was compared to evaluate whether BLG modification of nanoparticles limits premature drug release. Finally, the release of FBZ from the prepared nanoformulation into the targeted prostate cancer cells was investigated. Dynamic light scattering (DLS), scanning electron microscopy (SEM), transmission electron microscopy (TEM), Fourier transform–infrared (FTIR) spectroscopy, Brauer–Emmett–Teller (BET), differential scanning calorimetry (DSC), thermogravimetric analysis (TGA), and X-ray diffraction (XRD) measurements were used to characterize the formulations in terms of size, size distribution, zeta potential, morphology, chemical structure, specific surface area, glass transition temperature (T_g_), thermal stability, crystalline phases, and chemical composition. Moreover, the drug release was evaluated at different time intervals using reverse-phase high-performance liquid chromatography (RP-HPLC). The biological effects of the formulations were then studied in terms of cellular uptake, cell viability, cell proliferation, reactive oxygen species (ROS), and cell migration using laser scanning confocal microscopy, 3-[4,5-dimethylthiazol-2-yl]-2,5 diphenyl tetrazolium bromide (MTT), an IncuCyte live cell imaging system, a fluorescence microplate reader, and a fluorescence microscope, respectively.

## 2. Materials and Methods

### 2.1. Materials

Cetyltetramethylammonum bromide (CTAB), tetraethyl orthosilicate (TEOS), FBZ, phosphate-buffered saline (PBS), dimethyl sulfoxide (DMSO), tetraethyl orthosilicate (TEOS), D-α-tocopherol polyethylene glycol succinate (TPGS), carbonyldiimidazole (CDI), (3-Aminopropyl)triethoxysilane (APTES), sodium hydroxide (NaOH), ammonium hydroxide (NH_4_OH), 4′,6-diamidine-2′-phenylindole dihydrochloride (DAPI), phalloidin-FITC (fluorescein isothiocyanate), paraformaldehyde, bovine serum albumin (BSA), BLG, succinic anhydride, N-(3-Dimethylaminopropyl)-N′-ethylcarbodiimide hydrochloride (EDC.HCl), ethanol, NaOH, 2-(N-morpholino)ethanesulfonic acid (MES, low-moisture content ≥ 99%), Pluronic F127, dialysis tubes (2000 nominal molecular weight cutoff (NMWCO), benzoylated, with an average flat width of 32 mm), and triton X-100 were purchased from Merck (Castle Hill, Australia). medium RPMI-1640 (from Roswell Park Memorial Institute, Buffalo, NY, USA), trypsin-ethylenediaminetetraacetic acid (EDTA) (0.25%), fetal bovine serum (FBS), acetone, and formic acid (FA) were obtained from Thermo Fisher Scientific (Scoresby, Australia). MTT and 2′–7′dichlorofluorescin diacetate (DCFH-DA) were purchased from Abcam (Melbourne, Australia) and PromoKine (Promocell GmbH, Heidelberg, Germany), respectively. Cyanine5 NHS ester (Cy-5) was purchased from Tocris Bioscience, Victoria, Australia. Acetonitrile HPLC grade was prepared by RCI Labscan (Bangkok, Thailand). Deionized, double-distilled water was used throughout the experiments. PC-3 human prostate carcinoma cell line was a gift from Dr. Jennifer Gunter (Queensland University of Technology, Brisbane, Australia).

### 2.2. Synthesis of MCM-48 Nanoparticles

An amount of 0.5 g of CTAB was dissolved in NH_4_OH (107 mL, NH_3_ 2.8% *w*/*w*) and ethanol (43 mL) and stirred (1000 revolutions per minute (RPM)) until CTAB was solubilized. Next, 2.05 g of pluronic F127 was added to the solution and stirred until solubilized. TEOS (1930 µL) was added to the reaction medium and stirred (850 RPM) at room temperature (RT) for 60 s. The resulting mixture was kept under quiescent conditions for 24 h. Next, the obtained white suspension was centrifuged (12,000× *g* RPM, 10 min) to collect the formed MCM-48 nanoparticles. The nanoparticles were washed three times with distilled water and twice with ethanol. The nanoparticles were dried overnight in an oven (80 °C), then calcined at 550 °C for 5 h [16].

### 2.3. Surface Functionalization of MCM-48 with Amine (NH_2_) Group

MCM-48 nanoparticles (100 mg) were suspended in 10 mL of toluene and sonicated for 3 min. Next, 300 µL of APTES was added dropwise and stirred overnight (500 RPM, 37 °C). The suspension was then centrifuged at 12,000× *g* RPM for 20 min, and the pellets of nanoparticles were obtained, washed twice with ethanol and once with water (12,000 RPM, 3 min), and oven-dried at 60 °C.

### 2.4. Preparation of FBZ-MCM Nanoparticles

An amount of 40 mg of FBZ was dissolved in 15 mL of acetone as the loading solvent. Next, 160 mg of MCM-48-NH_2_ (MCM) nanoparticles was added to the reaction medium and stirred (300 RPM) at RT overnight. The organic solvent was removed using a rotary evaporator (Laborota 4000 HB/G1, Heidolph, Germany), and the formed FBZ-MCM nanoparticles were collected.

### 2.5. Synthesis of FBZ-MCM-BLG Nanoparticles

#### 2.5.1. Succinylation of BLG

BLG was succinylated according to the method described by Guillet-Nicolas et al. [17] Briefly, 200 mg of BLG was dissolved in 20 mL of PBS (pH 7.4), and 50 mg of succinic anhydride was added to the solution in 5 steps (10 mg each step) while stirring (300 RPM) for 1 h at RT. The pH of the solution was adjusted to 8 using 2 M NaOH and stirred for 20 min. The solution was then dialyzed using a benzoylated dialysis tube (1000 MWCO) against deionized water (4 °C, 24 h). The water was changed 5 times, and the sample-to-volume ratio was kept at 1:100. The solution was then freeze-dried.

#### 2.5.2. Grafting of BLG onto FBZ-MCM Nanoparticles

To graft BLG onto FBZ-MCM nanoparticles, 50 mg of BLG was dispersed in 8 mL of MES buffer (0.1 M, pH 6), and 6 mg of EDC was added under stirring to activate the carboxylic acid groups (20 min, RT). A total of 60 mg of FBZ-MCM nanoparticles was suspended in 2 mL of MES buffer (0.1 M, pH 6), added to the BLG/EDC mixture, and stirred for 2 h at RT. The formed nanoparticles were then centrifuged at 12,000× *g* RPM for 20 min and washed twice with MES buffer to remove the excess EDC and unreacted protein. The purified nanoparticles were freeze-dried under vacuum. To confirm that the BLG protein was conjugated to EDC and not physically adsorbed into the pores of the FBZ-MCM nanoparticles, a control reaction was carried out using the same procedures without adding FBZ to the reaction medium.

### 2.6. Cy-5 Grafting on MCM and MCM-BLG Nanoparticles

Cy-5 was attached by a covalent bond to MCM and MCM-BLG nanoparticles. Briefly, 30 mg of MCM and MCM-BLG nanoparticles were individually suspended in 3 mL of DMSO. Next, Cy-5 was solubilized in DMSO (3 mg/mL), and 1 mL of the solution was added to 3 mL of MCM and MCM-BLG suspensions and stirred (500 RPM) in a dark place at 4 °C for 24 h. Cy-5-loaded nanoparticles were then centrifuged at 10,000× *g* RPM for 5 min, washed three times with an ethanol:water mixture (75:25 *v*/*v*), and vacuum-dried.

### 2.7. Nanoparticles Characterization

#### 2.7.1. Dynamic Light Scattering (DLS) Measurements

DLS measurements were performed using a Malvern zetasizer (Malvern, UK). The suspension of 0.1% of MCM-48, MCM, FBZ-MCM, MCM-BLG, and FBZ-MCM-BLG nanoparticles was prepared in PBS, sonicated for 5 min, and loaded to the zetasizer instrument.

#### 2.7.2. Scanning Electron Microscopy (SEM) and Transmission Electron Microscopy (TEM)

MCM-48, MCM, FBZ-MCM, MCM-BLG, and FBZ-MCM-BLG nanoparticles were morphologically compared using Zeiss Sigma SEM and JEOL JEM-1400 TEM microscopes. For this purpose, the powder samples were mounted on a silicon wafer, coated with gold, and studied with a Zeiss Sigma SEM with a voltage rate of 2.5 kV. Additionally, to evaluate the nanoparticles using the TEM instrument, they were suspended in ethanol and deposited on a carbon-coated copper grid and then introduced to the TEM microscope at an accelerating voltage of 100 kV and a resolution of 0.25 nm.

#### 2.7.3. Thermogravimetric Analysis (TGA) and Differential Scanning Calorimetry (DSC) Measurements

The thermal decomposition behavior and kinetics of FBZ, MCM-48, MCM, FBZ-MCM, and MCM-BLG, compared to FBZ-MCM-BLG nanoparticles, were evaluated by TGA and DSC techniques using a NETZSCH 449 F3 Jupiter^®^ simultaneous thermal analyzer (STA). For this purpose, 5 mg of each formulation was heated in a predefined temperature range of 25–900 °C at a heating rate of 10 °C/min. The weight loss of the drug-loaded nanoparticles due to heating was measured, and the drug loading capacity was calculated by subtracting the final weight from the initial weight.

#### 2.7.4. X-ray Diffraction

FBZ, MCM-48, MCM, FBZ-MCM, and FBZ-MCM-BLG nanoparticles were characterized in terms of crystallographic structure, phase composition, and physical features using a Smartlab SE X-ray diffractometer with Cu Kα radiation (λ = 0.154 nm).

#### 2.7.5. Fourier Transform Infrared (FTIR) Measurements

The effects of protein conjugation to FBZ-MCM nanoparticles, compared to MCM-48, MCM, and FBZ-MCM nanoparticles, were studied using a Thermo Scientific Nicolet iD5 spectrometer (USA) and KBr pellets in the wavelength range of 400–4000 cm^−1^.

#### 2.7.6. Brunauer–Emmett–Teller (BET) Surface Area Analysis

The specific surface area, average pore diameter, and pore volume of MCM and FBZ-MCM nanoparticles were compared to those of the FBZ-MCM-BLG nanoparticles using a Micromeritics TriStar™ II 3020 system and the BET method.

#### 2.7.7. Release Study

The profiles of drug release from FBZ-MCM and FBZ-MCM-BLG nanoparticles were obtained at pH values of 1.2 and 6.8, respectively (corresponding to the pH of gastric and intestinal environments, respectively). Briefly, 4 mg of FBZ-MCM and FBZ-MCM-BLG nanoparticles containing 740 µg of FBZ was separately suspended in 5 mL of PBS at pH 1.2 and 6.8, respectively, and stirred at 37 °C (200 RPM). A volume of 100 µL of PBS was withdrawn from the suspension at time intervals of 0.25, 0.5, 1, 2, 4, 6, 8, 10, and 12 h and replaced with 100 µL of fresh PBS with the same pH. To solubilize the dispersed drug, 10 µL of DMSO was added to each sample, mixed, and centrifuged at 15,000× *g* RPM for 5 min. The drug concentration in the supernatant was measured using RP-HPLC chromatography (HPLC Series 1100, Agilent, Santa Clara, CA, USA) and a Kinetex C18 100A column (250 mm × 4.60 mm; 5 µm, Phenomenex, Ryde, Australia). The amount of FBZ released from FBZ-MCM and FBZ-MCM-BLG nanoparticles was calculated using a calibration plot for FBZ.

To achieve a calibration curve, 0.5 mg of FBZ was solubilized in acetonitrile/deionized water (50:50 *v*/*v*). The solution was diluted 6-fold with deionized water and centrifuged at 15,000× *g* RPM for 5 min. A 10 µL aliquot of the supernatant was then injected into the HPLC system, and the area under the curve (AUC) was calculated. The calibration curve was next obtained by plotting the concentration to AUC. Additionally, the drug release kinetics were also determined using zero-order, first-order, Higuchi, and Korsmeyer-Peppas mathematical models [18].

### 2.8. Qualitative Cellular Uptake Using Confocal Microscopy

All cell experiments were performed in accordance with the Research Ethics Committee of the Queensland University of Technology (number 2000000709). The cellular uptake of Cy5-MCM and Cy5-MCM-BLG into PC-3 cells was compared using the method described by Koohi Moftakhari Esfahani et al. [10] Briefly, PC-3 cells were seeded in RPMI-1640 media supplemented with 5% (*v*/*v*) FBS and 1% (*v*/*v*) penicillin/streptomycin (complete media) in a Cellvis 12-well plate or glass-bottom container at a concentration of 2 × 10^5^ cells/well. After 24 h incubation in 5% CO_2_ at 37 °C, the cells were treated with 100 µL of Cy5-MCM (100 µg/mL) and Cy5-MCM-BLG (100 µg/mL). The Cy5-MCM and Cy5-MCM-BLG formulations were prepared in the complete media and incubated in 5% CO_2_ at 37 °C for 4 h before mixing with the PC3 cells. The media were then removed, and the cells were washed three times with cold PBS (pH 7.4), fixed in paraformaldehyde (4% *v*/*v* in PBS) at RT for 20 min, and permeabilized for 30 min in Triton X-100 (0.2% *v*/*v* in PBS). To block nonspecific binding, the cells were treated with BSA (1% *v*/*v* in PBS) for 1 h at 4 °C. To stain filamentous actin, the cells were stained with phalloidin-FITC (40 mM) for 20 min and DAPI (14 mM) for 10 min, respectively. For control tests, cells were incubated only with the complete media without the addition of phalloidin-FITC and DAPI reagents. To prevent dehydration of cells, PBS was added, and the dish was covered with aluminium foil to protect the stains from light. The cells were then imaged using a laser scanning confocal microscope (Nikon Air Confocal, Adelaide, Australia) with a ×20 objective lens. Fluorophore excitations of DAPI and phalloidin-FITC were detected using 358–461 nm and 496–516 nm laser sources, respectively. Additionally, Cy-5 signals were recorded at excitation wavelengths of 646–670 nm.

### 2.9. Cell Viability

The effect of FBZ-MCM and FBZ-MCM-BLG nanoparticles, compared to FBZ, on the cell viability of human embryonic kidney (HEK)-293 and PC-3 cells was evaluated using an MTT assay [19]. Briefly, 8 × 10^3^ cells/well from each cell line were separately seeded in a Corning^®^ 96-well TC-treated microplate in the complete media and incubated in 5% CO_2_ at 37 °C for 48 h. Once the cells reached 80% confluency, the media were removed, and the cells were treated with 100 µL/well of the media containing FBZ, FBZ-MCM, and FBZ-MCM-BLG nanoparticles at drug concentrations of 6.25, 12.5, 25, 50, 100, 200, and 400 µM. Negative and positive control samples were prepared by treating the cells with the media only and the media containing sodium dodecyl sulfate (SDS) (10% *v*/*v* in water) + 0.1 M HCl, respectively. The media without cells were used as a background control. After incubation for 48 h at 37 °C in 5% CO_2_, the media were replaced with 100 μL/well of MTT solution (0.5 mg/mL in PBS) and incubated at 37 °C in 5% CO_2_. The MTT solution was discarded after 4 h, and 100 μL/well of DMSO was added, and the cells were incubated for a further 4 h at 37 °C in 5% CO_2_. The absorbance at 570 nm was then read, and the cell viability was measured using the following formula:(1)$$Cell\;viability=\frac{Absorbance_{sample}-Absorbance_{background}}{Absorbance_{negative\;control}-Absorbance_{background}}\times 100$$

### 2.10. Proliferation Assay

The effects of FBZ, FBZ-MCM, and FBZ-MCM-BLG nanoparticles on the proliferation of PC-3 cells were compared using an IncuCyte live cell imaging system (Essen Biosciences, Dandenong South, Vic, Australia). The cells were cultured in a 96-well plate in the complete media and incubated for 48 h in 5% CO_2_ at 37 °C to reach 80% confluency. The media were discarded, and the cells were treated with 100 µL/well of FBZ, FBZ-MCM, and FBZ-MCM-BLG nanoparticles at a drug concentration of 16.5, 33, and 66 µM. The cell confluency of the treated and untreated cells was then monitored every 10 h for 120 h.

### 2.11. Reactive Oxygen Species (ROS) Assay

The efficacy of FBZ, FBZ-MCM, and FBZ-MCM-BLG nanoparticles to produce intracellular ROS was measured based on the half maximal inhibitory concentration (IC_50_) of FBZ. For this purpose, 8 × 10^3^ PC-3 cells/well were seeded in a 96-well plate in the complete media to reach 70% confluency. Next, the media were removed, and the cells were treated with FBZ, FBZ-MCM, and FBZ-MCM-BLG nanoparticles at a drug IC_50_ concentration of 33 µM and incubated for 6 h at 37 °C in 5% CO_2_. The media were then discarded, and 100 µL of 20 µM DCFH-DA solution was added to the wells and incubated in the dark for 30 min at RT. The cells were rinsed three times with PBS, and the fluorescence intensity was measured using a fluorescence microplate reader at excitation and emission wavelengths of 485 and 520 nm, respectively.

### 2.12. Cell Migration

The effect of FBZ and FBZ-MCM nanoparticles, compared to FBZ-MCM-BLG nanoparticles, to inhibit cell invasion was evaluated using a cell migration assay. PC-3 cells were seeded in the complete media in a 6-well plate (CELLSTAR; Greiner Bio-One, Tokyo, Japan) and incubated in 5% CO_2_ at 37 °C for 48 h to reach 90–95% confluency. The cell monolayer was scratched with a pipette tip, washed with the medium to remove the cellular debris, and imaged at 0 h using an Olympus CKX41SF fluorescence microscope (Olympus, Tokyo, Japan). The media were removed, and the cells were treated with FBZ, FBZ-MCM, and FBZ-MCM-BLG nanoparticles at an IC_50_ of 33 µM. The cells incubated only with the media were regarded as the negative control. After 24 and 48 h incubation times, the scratches were imaged.

### 2.13. Statistical Analysis

GraphPad Prism software version 8.00 (GRAPH PAD Prism Software Inc., San Diego, CA, USA) was used for data analysis. The data concerning size, polydispersity index (PDI), drug loading efficiency, nitrogen (N_2_) adsorption–desorption isotherm analysis, cell viability, cell proliferation assay, and ROS assay were expressed as mean ± standard deviation (SD, *n* = 3). Statistical differences were determined using a one-way analysis of variance (ANOVA) test. Statistical analysis was performed using nonlinear regression analysis, and comparisons were made for IC_50_ values utilizing Tukey’s test.

## 3. Results and Discussion

### 3.1. Nanoparticle Characterization

#### 3.1.1. Dynamic Light Scattering (DLS) Measurements, Scanning Electron Microscopy (SEM), and Transmission Electron Microscopy (TEM)

Nanoparticle size has a fundamental effect on the efficiency of the loaded therapeutics. Small nanoparticles can internalize and transfect cells more efficiently than large nanoparticles. This results in a higher intracellular concentration of the loaded therapeutics [20]. Nanoparticles with a size below 300 nm can efficiently internalize into target cells and demonstrate their therapeutic effects [21]. In the current study, the size, PDI, and zeta potential of FBZ-MCM-BLG nanoparticles were measured and compared with MCM-48, MCM, and FBZ-MCM nanoparticles. The results demonstrated that the conjugation of BLG to FBZ-MCM caused an increase in the size of nanoparticles, compared to that of FBZ-MCM (233 ± 5 versus 369 ± 28 nm respectively), which confirms BLG conjugation into the particles (Figure 1A–C). However, FBZ-MCM-BLG nanoparticles were still sufficiently small to transfect the target cells. Additionally, as expected, MCM-48 (197 ± 1 nm), MCM (325 ± 9 nm), FBZ-MCM, and MCM-BLG (250 ± 2 nm) nanoparticles had different sizes, which indicates the conjugation of the amine group, BLG, and loading of FBZ into the nanoparticles. All formulations demonstrated PDI values below 0.4 (0.150 ± 0.014, 0.336 ± 0.018, 0.247 ± 0.006, 0.305 ± 0.024, and 0.298 ± 0.030 for MCM-48, MCM, FBZ-MCM, MCM-BLG, and FBZ-MCM-BLG, respectively), demonstrating that the formulations were monodisperse and homogenous [22]. In addition, the zeta potential for MCM-48, MCM, FBZ-MCM, MCM-BLG, and FBZ-MCM-BLG nanoparticles was found to be −29 ± 0.6, 20 ± 1.0, 17 ± 0.3, 30 ± 0.2, and 28 ± 0.4 mV, respectively. Further, TEM and SEM measurements confirmed that all nanoformulations were spherical, homogenous, and monodisperse (Figure 1D,E). Nanoparticles with high zeta potential >30 mV (either positive or negative) are more monodisperse than those with low zeta potential (<5 mV) and can easily aggregate [23]. In this study, the zeta potential of the nanoparticles was >5 mV and <−5 mV, which could inhibit their aggregation.

#### 3.1.2. Thermogravimetric Analysis (TGA) and Differential Scanning Calorimetry (DSC) Measurements

The thermal decomposition behavior and kinetics of FBZ-MCM-BLG nanoparticles were evaluated by TGA and DSC measurements and compared to those of the MCM-48, MCM, FBZ-MCM, and MCM-BLG nanoparticles. The TGA curve for FBZ demonstrated an initial weight loss at ~212 °C, followed by the complete decomposition of the drug at 710 °C. For MCM-48 nanoparticles, a 5 wt.% weight loss occurred after heating to 900 °C, which can be attributed to the evaporation of the water molecules adsorbed into the structure of the nanoparticles. MCM nanoparticles started to lose their weight at 82 °C, resulting in a 12 wt.% weight loss, which corresponds to the molecular weight of the amine group and evaporation of the water molecules adsorbed into the structure of the nanoparticles. The TGA measurements of the FBZ-MCM nanoparticles demonstrated a 2-step weight loss at 157 and 405 °C, respectively, and the total weight loss at 900 °C amounted to 30%. The TGA measurement for the MCM-BLG nanoparticles also demonstrated a 20 wt.% weight loss at 900 °C. FBZ-MCM-BLG nanoparticles demonstrated a weight loss at 148 °C. The total weight loss at 900 °C was 30%. Based on these results, the drug loading capacity for the FBZ-MCM and FBZ-MCM-BLG nanoparticles was ~19%, which is approximately 91% of the used drug (Figure 2A).

The DSC measurement of FBZ in the standard form demonstrated a glass transition temperature (Tg) at 190 °C (indicated by a baseline shift), followed by an exothermic peak at 220 °C, which corresponds to the drug melting point [24]. However, the DSC measurement of FBZ in FBZ-MCM and FBZ-MCM-BLG formulations demonstrated no melting peak, thus indicating a lack of phase transitions and a crystalline form of the drug. This result indicates that the drug transformed from the crystalline to the amorphous form in these formulations, which can improve its solubility and dissolution rate in aqueous media [9,25]. MCM-48, MCM, FBZ-MCM, MCM-BLG, and FBZ-MCM-BLG nanoparticles demonstrated no considerable peak in the temperature range of 50 to 350 °C, indicating that these formulations had excellent thermal stability. These results are in agreement with those of TGA, where the nanoparticles demonstrated high thermal stability in the tested temperature range (Figure 2B).

#### 3.1.3. X-ray Diffraction

To determine the effects of BLG conjugation on the crystallographic structure, phase composition, and physical features of the FBZ-MCM, MCM-48, MCM, and FBZ-MCM nanoparticles, XRD measurements were carried out. Additionally, XRD was used to determine the state of the loaded drug in these formulations. It has been demonstrated that porous nanocarriers could preserve the amorphous state of drugs in their pores and prevent their recrystallization [26]. The results of XRD analysis demonstrated that the drug in all nanoformulations retained a mesoporous structure with a 3D Ia3d cubic symmetry. In addition, regardless of their modification with the amine group and BLG, they retained the XRD peak positions ((211), (220), (321), (400), (420), (332), (422), and (431)), which confirmed the mesoporous state of the loaded drug (Figure 3) [27]. The results also demonstrated that FBZ had a non-crystalline state inside the pores of both MCM and BLG-MCM nanoparticles. FBZ demonstrated multiple XRD peaks at 2θ of 10, 17, 24.7, and 33.5, which were due to its crystalline nature. These peaks were not seen in the FBZ-MCM and FBZ-MCM-BLG formulations, thus confirming the amorphization of FBZ inside the pores of the nanoparticles. However, the BLG-MCM carrier was more potent than MCM to increase the amorphous state of the drug (Figure 3). The amorphous state of the loaded FBZ was also confirmed by DSC measurements (Figure 2B).

#### 3.1.4. Fourier Transform Infrared (FTIR) Measurements

The chemical composition of FBZ, MCM-48, MCM, FBZ-MCM, MCM-BLG, and FBZ-MCM-BLG nanoparticles was investigated by FTIR (Figure 4). Additionally, FTIR was used to confirm drug loading into the nanoparticles (Figure 4). The characteristic bands at 1052 and 786 cm^−1^ were attributed to MCM-48 nanoparticles, confirming the synthesis of the nanoparticles [28]. The vibration bands at 1622, 736, and 688 cm^−1^ in the FBZ-MCM and FBZ-MCM-BLG formulations were attributed to the physically adsorbed FBZ in the nanoparticle pores (Figure 4).

#### 3.1.5. Brunauer–Emmett–Teller (BET) Surface Area Analysis

The pore size and BET surface area of MCM and FBZ-MCM nanoparticles were measured using N_2_ adsorption–desorption isotherm analysis. In the BET curves, type IV International Union of Pure and Applied Chemistry (IUPAC) isotherms were observed for all nanoformulations [29]. These isotherms are distinctive to MSNs [30], thus indicating that MCM and FBZ-MCM nanoparticles, such as MSNs-based particles, were successfully synthesized. The BET surface area for FBZ-MCM nanoparticles was 112 m^2^/g, which is less than that of MCM nanoparticles (264 m^2^/g). This could be attributed to the loading of FBZ molecules into the nanoparticles (Figure 2C). The loading of the drug also causes a significant decrease in the pore volume of FBZ-MCM nanoparticles as compared to that of MCM nanoparticles (0.04 and 1.75 cm^3^/g, respectively; Figure 2D). Furthermore, the grafting of BLG moieties into the FBZ-MCM nanoparticles led to full coverage of the particle’s outer surface and pores, resulting in no BJH pore size and pore volume. The BJH pore size for MCM and FBZ-MCM nanoparticles was equal to 1.75 and 1.8 nm, respectively.

#### 3.1.6. Release Study

The drug release from FBZ-MCM-BLG was evaluated and measured at two pH values of 1.2 and 6.8, corresponding to the pH of the gastric and intestinal environments, respectively. FBZ was released from FBZ-MCM nanoparticles in a biphasic manner, which was initiated by a burst release and followed by a sustained release until the end of the measurement. Proportions of 63 and 44% of the drug release from FBZ-MCM nanoparticles occurred in the first 30 min of the study at pH 1.2 and 6.8, respectively. After 30 min, the drug release continued in a sustained manner for 12 h until the end of the study, at which point 95 and 74% of the loaded drug was released at pH 1.2 and 6.8, respectively. The higher drug release at pH 1.2 as compared to that at pH 6.8 could be attributed to the effect of pH on the charge of the drug and MCM particles. At pH 1.2, both FBZ and MCM have a positive charge, and electrostatic repulsion occurs between the drug and the carrier, leading to the release of FBZ from the nanoformulation [10]. The pattern of drug release from MCM-BLG nanoparticles was similar to that from MCM nanoparticles. Biphasic drug release was observed at both pH 1.2 and 6.8. However, in the first 30 min of the measurements, a lesser amount (17 and 15% at pH 1.2 and 6.8, respectively) of the drug was released from MCM-BLG nanoparticles when compared to that from MCM nanoparticles. Proportions of 53% and 38% of the loaded drug were released from FBZ-MCM-BLG at pH 1.2 and 6.8, respectively, whereas 63% and 44% of the loaded drug was released from FBZ-MCM at the same pH values. After the initial burst release, the drug release from MCM-BLG nanoparticles continued in a sustained manner for 12 h, where 79 and 62% of the loaded drug was released at pH 1.2 and 6.8, respectively (Figure 5). These results indicate that both MCM and MCM-BLG effectively controlled the release of FBZ for 12 h under acidic gastric and natural intestinal conditions.

In the zero-order kinetic model, the release of drugs from the polymer matrix is not dependent on the drug concentration, meaning that the amount of drug release per unit of time is constant throughout an experiment. In the first-order kinetic model, the drug release rate is dependent on the drug concentration; therefore, the amount of drug release is reduced with time [31]. In the Higuchi kinetic model, there is a direct relationship between the cumulative drug release and the square root of time [32]. In this model, diffusion is the predominant mechanism of drug release [33,34,35], whereas in the Korsmeyer-Peppas model, the rate of drug release is determined by the diffusion and swelling rate [36,37]. In this study, according to the results, the drug release profiles from FBZ-MCM and FBZ-MCM-BLG at both pH 1.2 and 6.8 followed the Higuchi model, with correlation coefficient (R^2^) values of 0.7357 and 0.7865 for FBZ-MCM at pH 1.2 and 6.8, respectively, and R^2^ values of 0.7408 and 0.773 for FBZ-MCM-BLG at pH 1.2 and 6.8, respectively (Appendix A).

#### 3.1.7. Qualitative Cellular Uptake Using Confocal Microscopy

The cellular uptake of MCM-BLG nanoparticles was qualitatively measured and compared to that of MCM using a confocal microscope. Figure 6A demonstrates the fluorescence images of these nanoparticles after incubation with PC-3 cells for 4 h. As the figure shows, fluorescence intensity was observed at the peripheral of the cells, which were incubated with Cy5-MCM and Cy5-MCM-BLG nanoparticles (Figure 6A). This result indicates that there was no difference in the rate of cellular uptake between the two nanoparticles.

#### 3.1.8. Cell Viability

The effects of FBZ-MCM-BLG and FBZ-MCM nanoparticles on HEK-293 and PC-3 cell viability were compared to those of FBZ alone using an MTT assay. It was found that MCM and MCM-BLG at concentrations of 125 µg/mL were safe and non-toxic for both cell lines. All other formulations (FBZ, FBZ-MCM, and FBZ-MCM-BLG) caused a reduction in PC-3 cell viability in a dose-dependent manner, where the cell viability decreased by increasing the drug concentration (Figure 6B). However, these formulations (FBZ, FBZ-MCM, and FBZ-MCM-BLG) were safe and non-toxic against HEK-293 cells (Appendix A). Both FBZ-MCM and FBZ-MCM-BLG were found to increase the cytotoxic effects of the drug against prostate cancer PC-3 cells (IC_50_: 33, 11, and 6 µM for FBZ, FBZ-MCM, and FBZ-MCM-BLG nanoparticles, respectively). MCM-BLG nanoparticles were found to be more effective in decreasing the cell viability of PC-3 cells than MCM nanoparticles by 1.8-fold. The higher cytotoxicity effects of FBZ-MCM-BLG nanoparticles as compared to those of the FBZ-MCM nanoparticles could be attributed to the drug release pattern of FBZ-MCM-BLG nanoparticles, as these particles could release the loaded drug over a longer time. Moreover, these nanoparticles demonstrated higher cellular uptake efficiency, resulting in higher pharmacological activity.

#### 3.1.9. Proliferation Assay

The effects of FBZ-MCM and FBZ-MCM-BLG nanoparticles on the proliferation of PC-3 cells were compared to those of the standard drug using an IncuCyte live cell imaging measurement. As demonstrated by Figure 6, FBZ, FBZ-MCM, and FBZ-MCM-BLG nanoparticles caused an inhibition in the proliferation of PC-3 cells that increased with an increase in the drug concentration (Figure 6C). Additionally, both MCM and MCM-BLG nanoparticles increased the cytotoxic effects of FBZ, resulting in less proliferation of PC-3 cells compared to that of FBZ alone. However, MCM-BLG nanoparticles were more potent in inhibiting cell proliferation when compared to MCM nanoparticles. These results are in agreement with the results obtained from the cell viability measurements.

#### 3.1.10. Reactive Oxygen Species (ROS) Assay

The effects of FBZ, FBZ-MCM, and FBZ-MCM-BLG nanoparticles on the production of intracellular ROS were measured. Most chemotherapeutics are able to elevate the intracellular concentrations of ROS [38]. The overproduction of ROS inhibits the efflux pumps in multidrug-resistant (MDR) cancer cells and reverses the resistance of these cells to chemotherapeutics, resulting in more MDR cell death [10]. In the present study, the results of the ROS assay demonstrated that FBZ, FBZ-MCM, and FBZ-MCM-BLG nanoparticles increased the production of intracellular ROS in PC-3 cells (Figure 6D). The loading of FBZ into the MCM and MCM-BLG nanoparticles caused an increase in the efficacy of FBZ to produce ROS by 1.3- and 1.6-fold, respectively, indicating that FBZ-MCM-BLG nanoparticles were the most potent formulations to produce intracellular ROS. The higher potency of FBZ-MCM-BLG nanoparticles to produce higher amounts of intracellular ROS concentration, when compared to that of FBZ-MCM nanoparticles, could be attributed to the pattern of drug release (drug release over a longer period of time) and the high cellular uptake efficiency of FBZ-MCM-BLG nanoparticles. Previous studies [39,40] have demonstrated that MSNs by themselves cause an increase in the production of intracellular ROS, whereas some other researchers [41] have shown that MSNs can be used as a drug carrier to remove intracellular ROS.

#### 3.1.11. Cell Migration

Cancer cells can invade the surrounding tissues by cell migration [42]. Thus, the effects of FBZ-MCM and FBZ-MCM-BLG nanoparticles in inhibiting PC-3 cell migration were measured using a cell migration assay and compared to those of FBZ alone. The results showed that FBZ, FBZ-MCM, and FBZ-MCM-BLG nanoparticles inhibit cell migration in a time-dependent manner when compared to the control group (Figure 7). However, both FBZ-MCM and FBZ-MCM-BLG nanoformulations were more potent than FBZ alone in inhibiting PC-3 cell migration (Figure 7B). The results also showed that FBZ-MCM-BLG nanoparticles were more efficient than FBZ and FBZ-MCM formulations in inhibiting cell migration. These results are in line with the results of cell viability measurement, in which FBZ-MCM-BLG nanoparticles were found to be more efficient than FBZ and FBZ-MCM nanoparticles in decreasing cell viability.

## 4. Conclusions

In this study, we developed FBZ-loaded mesoporous nanoparticles to improve their oral delivery and anticancer effects as a candidate anticancer drug. A new pH-responsive nanoformulation of FBZ was developed using amine-functionalized MCM-48 nanoparticles and succinylated BLG. The nanoformulation improved the profile of drug release and the anticancer effects of FBZ. The nanoparticles demonstrated high drug loading capacity, with ~91% of the used drug loaded into the nanoparticles. The conjugation of succinylated BLG into MCM nanoparticles increased the efficacy of the nanoparticles to control drug release. Additionally, the conjugation of succinylated BLG into the nanoparticles caused a significant increase in the cytotoxicity and cellular uptake of FBZ against human prostate cancer PC-3 cells. In addition, these nanoparticles, compared to FBZ and FBZ-MCM nanoparticles, were more potent in inhibiting cancer cell migration and invasion and producing intracellular ROS by 1.6- and 1.2-fold, respectively. The prepared FBZ nanoparticles with improved FBZ delivery to PC-3 cells may be useful against prostate cancer; however, further in vivo investigations are warranted.

## Figures and Tables

**Figure 1 pharmaceutics-14-00884-f001:**
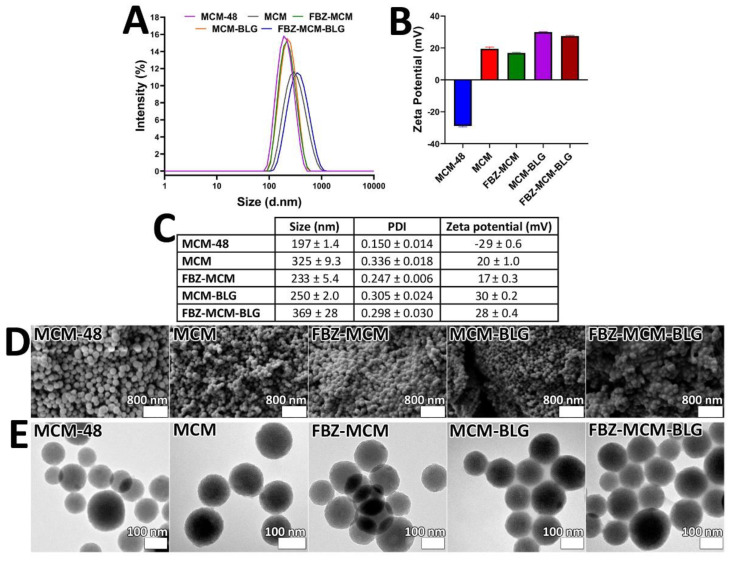
(**A**) Size diagrams of MCM-48, MCM, FBZ-MCM, MCM-BLG, and FBZ-MCM-BLG nanoparticles. (**B**) Zeta potential of MCM-48, MCM, FBZ-MCM, MCM-BLG, and FBZ-MCM-BLG nanoparticles. (**C**) Size, polydispersity index (PDI), and zeta potential of the formulations. (**D**) Scanning electron microscopy (SEM) images and (**E**) transmission electron microscopy (TEM) images of MCM-48, MCM, FBZ-MCM, MCM-BLG, and FBZ-MCM-BLG nanoparticles. As the figure demonstrates, spherical, monodisperse, and homogenous nanoparticles were formed.

**Figure 2 pharmaceutics-14-00884-f002:**
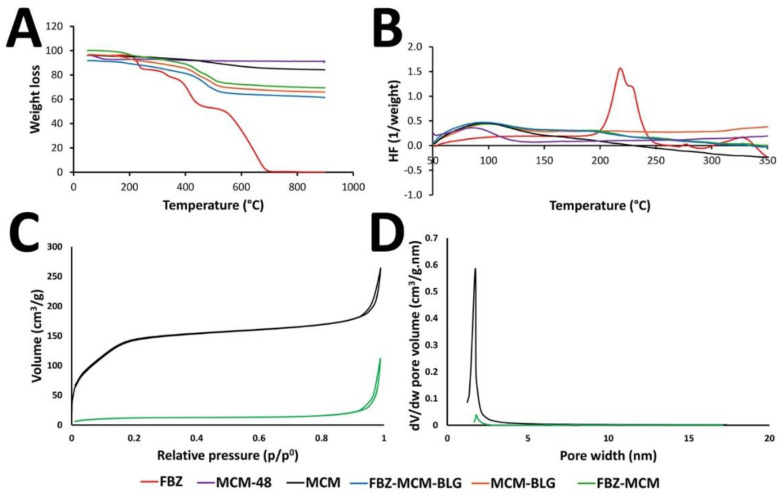
(**A**) TGA thermogram of FBZ, MCM-48, MCM, FBZ-MCM, MCM-BLG, and FBZ-MCM-BLG nanoparticles; (**B**) DSC thermogram of FBZ, MCM-48, MCM, FBZ-MCM, MCM-BLG, and FBZ-MCM-BLG nanoparticles; (**C**) N2 adsorption-desorption isotherms; and (**D**) pore size distributions of MCM and FBZ-MCM nanoparticles using the BET method.

**Figure 3 pharmaceutics-14-00884-f003:**
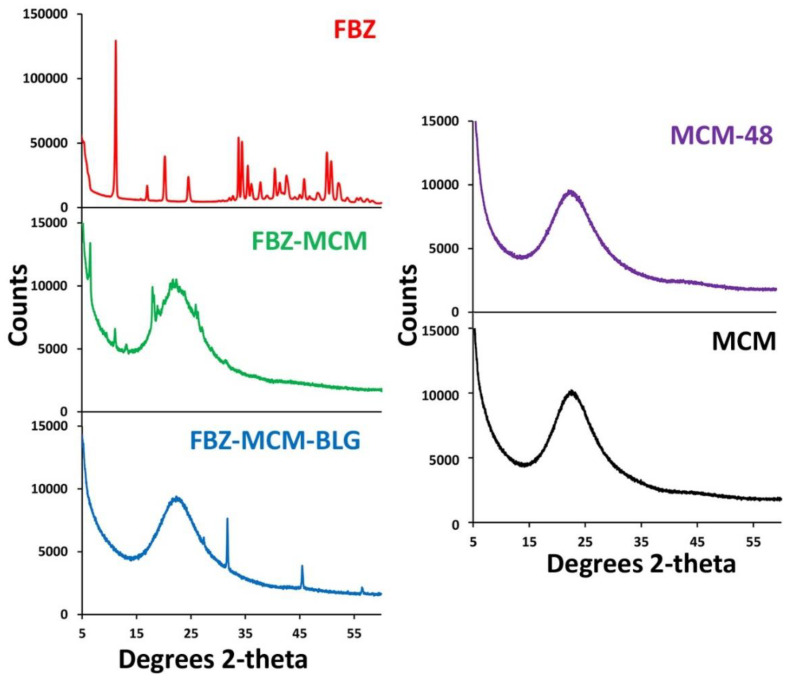
XRD pattern of the synthesized MCM-48, MCM, FBZ-MCM, and FBZ-MCM-BLG nanoparticles.

**Figure 4 pharmaceutics-14-00884-f004:**
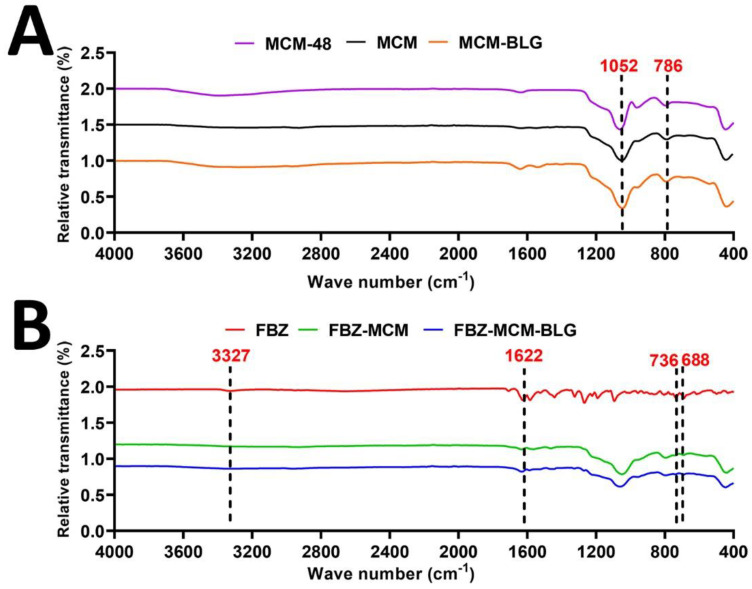
Fourier transform infrared spectroscopy (FTIR) spectra of (**A**) MCM-48, MCM, and MCM-BLG; and (**B**) FBZ, FBZ-MCM, and FBZ-MCM-BLG nanoparticles. The peaks at 1622, 736, and 688 cm^−1^ regions in the spectra of FBZ, FBZ-MCM, and FBZ-MCM-BLG nanoparticles prove that (i) FBZ was loaded into these nanoparticles and (ii) the chemical structure of the drug remained intact, meaning that FBZ was physically loaded into these NPs.

**Figure 5 pharmaceutics-14-00884-f005:**
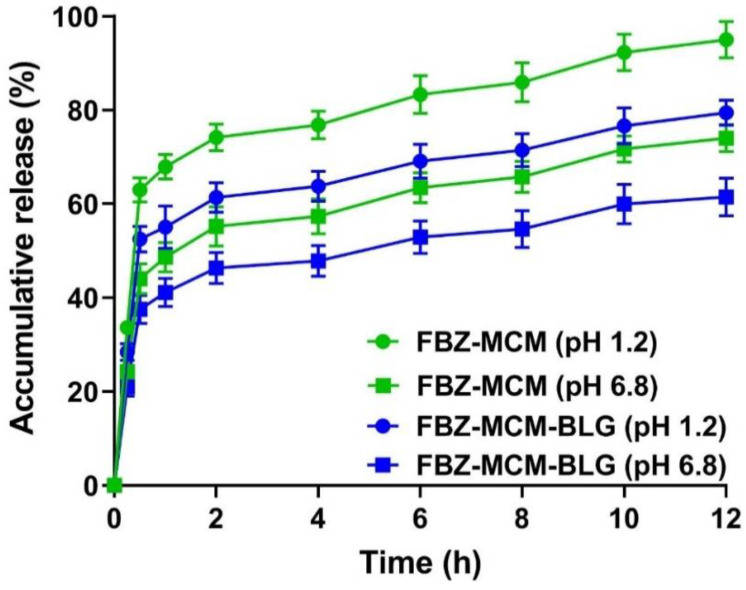
Drug release pattern from FBZ-MCM and FBZ-MCM-BLG nanoparticles, calculated at pH 1.2 and 6.8. One-way analysis of variance (ANOVA) and F tests were used for statistical analysis of the data. Additionally, the values of the slope (accumulative release/hour) for FBZ-MCM and FBZ-MCM-BLG at pH 1.2 were 5.0 and 4.2, respectively, whereas these values at pH 6.8 were 4.2 and 3.4, respectively. The data are expressed as mean ± SD (*n* = 3).

**Figure 6 pharmaceutics-14-00884-f006:**
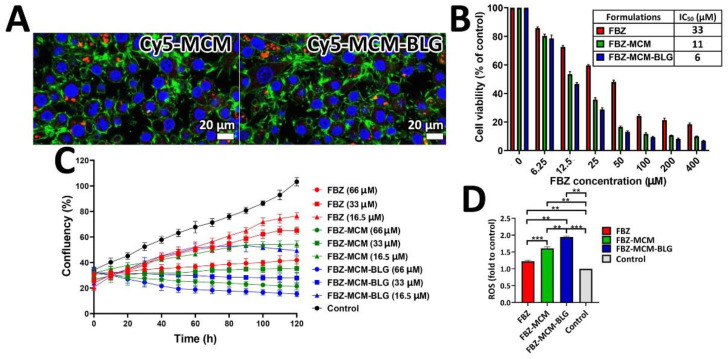
(**A**) Cellular uptake images of PC-3 cells after 4 h incubation with Cy5-MCM and Cy5-MCM-BLG nanoparticles. As the images show, higher fluorescence intensity was observed in the cells incubated with MCM-BLG nanoparticles. (**B**) Viability effects of FBZ-MCM and FBZ-MCM-BLG nanoparticles compared to FBZ on prostate cancer PC-3 cells after 48 h incubation. The data are expressed as mean ± SD (*n* = 3). (**C**) Cell proliferation effects of FBZ, FBZ-MCM, and FBZ-MCM-BLG nanoparticles on PC-3 cells over 5 days. (**D**) ROS generation in PC-3 cells after 6 h incubation with FBZ, FBZ-MCM, and FBZ-MCM-BLG nanoparticles. The results are expressed as mean ± SD from three independent experiments and were analyzed using a t-test: ** *p* < 0.01, *** *p* < 0.001.

**Figure 7 pharmaceutics-14-00884-f007:**
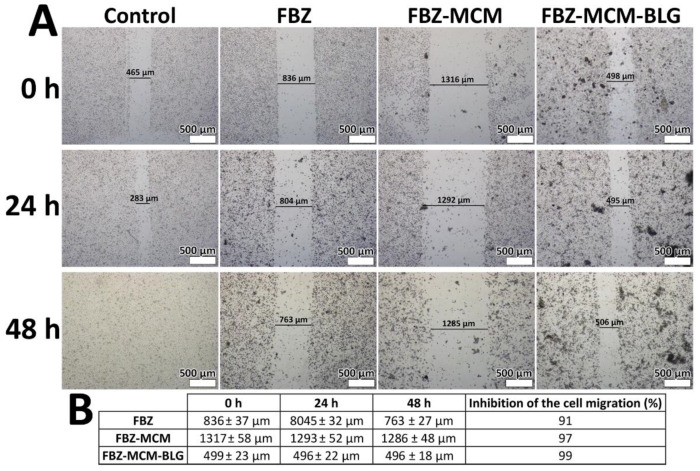
(**A**,**B**) Effects of FBZ, FBZ-MCM, and FBZ-MCM-BLG nanoparticles on the migration and invasion of prostate cancer PC-3 cells before (0 h), 24 h after, and 48 h after cell treatment (×40 mag). Compared to the control group, FBZ, FBZ-MCM, and FBZ-MCM-BLG restrained cell migration, confirming the efficacy of FBZ, FBZ-MCM, and FBZ-MCM-BLG in inhibiting prostate cancer cell metastasis and invasion.

## Data Availability

Not applicable.

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
