# Peer review of "β-Lactoglobulin-Modified Mesoporous Silica Nanoparticles: A Promising Carrier for the Targeted Delivery of Fenbendazole into Prostate Cancer Cells"

_pharmaceutics, 2022, doi:10.3390/pharmaceutics14040884_

Round 1

Reviewer 1 Report

In their article entitled " β-lactoglobulin-Modified Mesoporous Silica Nanoparticles: A Promising Carrier for the Targeted Delivery of Fenbendazole into Prostate Cancer Cells", authors demonstrated that their nano-objects has improved fenbendazole delivery to prostate cancer cells. The manuscript is well written. It proposes a very interesting transdisciplinary study starting from an exhaustive physico-chemical characterization of the nano-object to the biological proof of concept. You will find below my comments :

1. Make sure that abbreviations are defined when they are first used. Example : PDI 

2. Take into account the significant digits when reporting numerical values. Example (line 332): 368.8 +/- 28.3 nm can be rewritten as 368 +/- 28 nm

3. Section 3.1.6: Could you indicate the values of the two slopes (Accumulative release per time unit) for the two phases.

4. Figure 5: "Cumulative release" seems to me more appropriate than "Accumulative release". I think you can go further in the interpretation of these experimental data by fitting them with an adapted mathematical model. You could be inspired by the following manuscripts: Dash, S. et al. (2010), Acta Pol Pharm, 67(3), 217-223. or Mircioiu, C. et al. (2019), Pharmaceutics, 11(3), 140.

5. Section 3.1.7. You modify your nano-object by adding a dye. To what extent could the addition of this dye modify the internalization of the nano-object (internalization pathway, ...)? Do you have an idea of the relative size of this Cy5 dye compared to the size of your nano-object ? 

6. Figure 6: Could you add a scale bar on the images 6A to specify the magnitude ? 

7. Figure 6A: Do you observe a preferential intracellular localization of your nano-objects? (nucleus, perinuclear space, ...)

8. Section 3.1.8. You indicate IC50 values based on your MTT assay. I suppose that these were determined on the basis of a fit from a mathematical model since you do not report the corresponding experimental points in figure 6B. Could you please specify the model used in the Material & Method section?

9. In general: You often present results showing the effect of FBZ, FBZ-MCM and FBZ-MCM-BLG, as in Figure 7. Have you done additional controls MCM alone, BLG alone? This would allow to define if the data you reported with FBZ-MCM or FBZ-MCM-BLG result from a synergistic or additive effect. 

10. Section 3.1.10 & Figure 6D: How do you explain the higher ROS production with FBZ-MCM compared to FBZ alone? Is it related to a higher amount of FBZ released or to an effect of silica nanoparticles? Indeed, it has been shown that the incubation of metallic nanoparticles could induce oxidative stress (cfr. doi.org/10.1088/1361-6560/ab9159; doi.org/10.3390/cancers12082021). Could you clarify this point in section 3.1.10 ?

11. In order to have a global view of your work and to better understand the link between chemical mediators (ROS) and the biological endpoints you report (migration, cell proliferation), I strongly encourage you to quantify DNA damage in your system through a measurement of y-H2AX or 53BP1 foci y immunofluorescence, as defined by the guidelines (doi.org/10.1093/narcan/zcab046). Indeed, recent papers show that FBZ seems to act as a moderate microtubule destabilizing agent, which supports this request to investigate DNA damage (10.1038/s41598-018-30158-6)

Very nice work !

Author Response

Please read the attached file 

Reviewer 2 Report

The manuscript “β-lactoglobulin-Modified Mesoporous Silica Nanoparticles: A  Promising Carrier for the Targeted Delivery of Fenbendazole into Prostate Cancer Cells” is interesting for delivery of Fenbendazole into Prostate cancer cells. The authors presented well the research studies and support as proof-of-concept for this formulation and may be acceptable. However, in vivo studies will be proof further the significant enhancement of the investigation.

Reviewer 3 Report

Sustained drug release is an especially concerned subject. This study developed the FBZ loaded mesoporous nanoparticles to improve the oral delivery and anticancer effects as a candidate anticancer drug. The MCM-48 mesoporous nanoparticles were specially modified by succinylated BLG to enhance the cytotoxicity and cellular uptake of FBZ against human prostate cancer PC-3 cells. The prepared nanoparticles were well characterized, and some biological effects of the prepared were detected. The obtained results are useful to some extent for the study of sustained-release drugs.

(1) The effect of the prepared nano-particles on normal cells should be concerned also.

(2) Is the cell phagocytic activity the same before and after the fluorescent labeling of the nanoparticles?

(3) In Figure 6, “As the figure shows, high fluorescence intensity was observed from the cells that were incubated with Cy5-MCM-BLG nanoparticles, when compared to that observed from the cells incubated with Cy5-MCM nanoparticles (Figure 6A).” , This description does not clearly reflect results of the Figure 6. In fact, Figure 6 reflects neither the nanoparticles being englobed by the cells, and, nor the difference between Cy5-MCM 487 and Cy5-MCM-BLG.

(4) Line 365, 900℃should be checked.

(5) line 367-368, the loss weight at 80℃ was attributed to the molecular weight of the amine group. Why the loss weight here is not attributed to loss water.

Author Response

Read the attached file 

Round 2

Reviewer 1 Report

There are still minor coments :

  • Page 12, you added “Also, the values of the slope (accumulative release/time unit) for FBZ-MCM and FBZ-MCM-BLG at pH 1.2 were 5.0 and 4.2, respectively, while these values at pH 6.8 were 4.2 and 3.4, respectively.” Could you replace "time unit" by hours since I assume that the reported values are relative accumulative release per hour. 
  • Related to my previous comment #10 :  Your explanation must be mentioned somewhere in the manuscript. I find your answer interesting and so this can be added with some precautions. The oxidative stress that you observe could therefore come from the incubation of the nano-objects as observed in several studies (doi.org/10.1088/1361-6560/ab9159; doi.org/10.3390/cancers12082021; 10.1016/j.biomaterials.2010.04.055) but other studies tend to show that they can also be used as a carrier to remove ROS (doi.org/10.1016/j.chempr.2019.05.023 ; 10.1016/j.actbio.2020.12.029 )

  • Related to my previous comment #11 : I understand that the purpose of this paper is to provide a proof of concept of the benefit of your nano-object. Nevertheless, if you want to go further in the understanding of the impact of this one at the biological level, I recommend you to investigate the DNA damage pathway (quantification of the induction and repair of these). 

Author Response

Please read attached document 
